# Preparation of Proline-Modified UIO−66 Nanomaterials and Investigation of Their Potential in Lipase Immobilization

Xiaoxiao Dong [1,2,†], Chengnan Zhang [3,†], Prasanna J. Patil [1,2] ⓘ, Weiwei Li [1,2] ⓘ and Xiuting Li [1,2,*] ⓘ

1   Key Laboratory of Geriatric Nutrition and Health, Beijing Technology and Business University, Ministry of Education, Beijing 100048, China; dongxiaoxiao_2475@163.com (X.D.); prasanna.j.patil57@st.btbu.edu.cn (P.J.P.); liweiwei.0304@163.com (W.L.)
2   Key Laboratory of Brewing Microbiome and Enzymatic Molecular Engineering, China General Chamber of Commerce, Beijing 100048, China
3   Department of Exercise Biochemistry, Exercise Science School, Beijing Sport University, Beijing 100084, China; zhangcn@btbu.edu.cn
*   Correspondence: lixt@btbu.edu.cn; Tel.: +86-10-68985342
†   These authors contributed equally to this work.

**Abstract:** Metal–organic frameworks (MOFs) are regarded as excellent carriers for immobilized enzymes due to their substantial specific surface area, high porosity, and easily tunable pore size. Nevertheless, the use of UIO−66 material is significantly limited in immobilized enzymes due to the absence of active functional groups on its surface. This study comprised the synthesis of UIO−66 and subsequent modification of the proline (Pro) on UIO−66 through post-synthetic modification. UIO−66 and UIO−66/Pro crystals were employed as matrices to immobilize *Rhizopus oryzae* lipase (ROL). The contact angle demonstrated that the introduction of Pro onto UIO−66 resulted in favorable conformational changes in the structure of ROL. The immobilized enzyme ROL@UIO−66/Pro, produced via the covalent-bonding method, exhibited greater activity (0.064715 U/mg (about 1.73 times that of the free enzyme)) and stability in the ester hydrolysis reaction. The immobilized enzymes ROL@UIO−66 (131.193 mM) and ROL@UIO−66/Pro (121.367 mM), which were synthesized using the covalent-bonding approach, exhibited a lower $K_m$ and higher substrate affinity compared to the immobilized enzyme ROL@UIO−66/Pro (24.033 mM) produced via the adsorption method. This lays a solid foundation for the industrialization of immobilized enzymes.

**Keywords:** *Rhizopus oryzae* lipase; UIO−66; proline modification; adsorption; covalent bonding





## 1. Introduction

Among the most popular enzymes, lipases (EC 3.1.1.3) have enormous biocatalytic potential in numerous areas of industrial microbiology and biotechnology and are today considered an integral part of a wide variety of industrial processes [1,2]. Lipase-catalyzed processes encompass esterification, inter-esterification, hydrolysis, alcholysis, acidolysis, and aminolysis [3]. Lipases that exhibit selectivity towards the 1,3-position are crucial for the synthesis of 1,3-diacylglycerol and structured lipids. *Rhizopus oryzae* lipase (ROL) is one of the commercially available lipases, and exhibits 1,3 regional specificity, making it suitable for application in the food, medicinal, and energy sectors [4–6]. ROL has proven effective in transforming microalgal lipids into fatty acid methyl esters (FAME) in an enzymatic methanolysis reaction, leading to the synthesis of biodiesel [7,8]. Furthermore, 2-monoacylglycerol synthesized by ROL has the potential to enhance the cost efficiency of biodiesel refining and is employed as an emulsifying agent in the food and pharmaceutical sectors [9–11]. Nevertheless, despite the numerous benefits of lipases generated from *Rhizopus oryzae*, challenges still hinder their practical implementation in industrial settings. The synthesis of industrial enzymes must take into account the crucial issues of the operational stability of the biological catalyst and the economic competitiveness of

traditional methods due to the expensive nature of enzymes and the economic competition they face [12]. One method to address these limitations of lipase is immobilizing it onto a solid substrate. This allows for continuous operation, preventing contamination of enzyme products and enhancing the stability and economic benefits of the enzyme and process. In some cases, it can also increase enzyme activity [13–15]. It has been shown that human milk fat substitute (HMFS) can be synthesized by acid digestion of lard and fish oil by immobilized heterologous ROL with an efficiency similar to that of commercial lipases [4]. Continuous esterification reaction of fusel oil by immobilizing ROL on mesoporous poly particles demonstrates excellent catalytic performance and stability for the production of biolubricants [16].

Metal–organic frameworks (MOFs) are a type of structured crystalline solid that is created by linking metal ions or metal clusters with organic linkers [17,18]. In recent years, MOF materials have exhibited substantial promise in numerous areas of research, including catalysis, drug delivery, biosensing, gas storage and separation, food science, biodiesel synthesis, and many more [1,2]. Due to its substantial specific surface area, straightforward synthesis and modification process, and favorable physicochemical features, it has been regarded as a viable material for lipase immobilization [19,20]. UIO−66 is a microporous substance composed of zirconia clusters. The material possesses a crystal structure that is face-centered cubic and exhibits exceptional physical, chemical, and hydrothermal stability. UIO−66 has exceptional stability throughout a broad spectrum of temperatures and acid–base conditions, distinguishing it from other MOFs [21–24]. According to its face-centered cubic (fcu) structure, UIO−66 has a total of 12 connectors. Out of these, 4 connectors are linked as hydroxides to create a stable hydroxylated state [25]. However, there is still a noticeable scarcity of active groups specifically designed for immobilized lipases. As a result, adsorption alone is frequently employed as a substitute [26–28]. Nevertheless, the adsorption technique just binds the carrier to the enzyme via non-specific and feeble interactions, hence facilitating the enzyme's detachment. Furthermore, if the rate at which the enzyme is immobilized is greater than the rate at which it can diffuse through the pore, the enzyme tends to clump together, causing a steric problem. This leads to a resistance in the transfer of mass between the substrate and the enzyme, resulting in a significant decrease in enzyme activity [29,30]. As a result, application of the immobilized enzyme of UIO−66 is limited.

Presently, post-synthetic modification of MOFs is commonly employed to add functional groups in MOFs. For example, Gou et al. modified the long alkyl chains of UIO−66-NH2 to enhance the thin-film nanocomposite (TFN) membrane's methanol permeability [31]. In order to construct membranes, Zhou et al. added amine groups to polycrystalline ZIF-67. The MOF membranes treated with amines exhibited enhanced adsorption capacity [32]. Rani et al. chemically modified the Cu-MOF surface by functionalizing it with three amino acids [33]. Studies have shown that incorporating Pro into UIO−66 by the hydrothermal method leads to beneficial alterations in the structure. The modified substance is subsequently employed in the production of the immobilized enzyme using the adsorption technique, resulting in enhanced activity and stability [34]. Nevertheless, the impact of incorporating amino acids into UIO−66 to alter its characteristics and facilitate the immobilization of ROL through various techniques is still not well understood.

The present study involved the synthesis of Pro-modified UIO−66 by the application of covalent-bonding and adsorption methods for the purpose of immobilizing ROL. The study investigated the morphological and structural differences among UIO−66, UIO−66/Pro, and immobilized ROL (ROL@UIO−66 and ROL@UIO−66/Pro) using UIO−66 and UIO−66/Pro materials. In addition, we compare the temperature stability, pH stability, reusability, and kinetic analysis of free ROL enzymatic activities with those of ROL@UIO−66 and ROL@UIO−66/Pro. Our work offers additional insights for immobilizing lipases using different immobilization techniques based on post-synthesis modified MOFs. Consequently, we anticipate that this research will assist researchers and industrialists in creating more effective MOF-lipase biocomposites.

## 2. Results and Discussion

### 2.1. Synthesis and Characterization of UIO−66, UIO−66/Pro, ROL@ UIO−66, and ROL@ UIO−66/Pro

The UIO−66 material was synthesized by combining zirconium chloride and terephthalic acid in a 1:1 molar ratio [Figure 1a]. The precursor for the UIO−66 material was generated by mixing UIO−66 and Pro in a 2:5 molar ratio. ROL@UIO−66 and ROL@UIO−66/Pro were synthesized by incorporating UIO−66 and UIO−66/Pro carriers into an ROL solution [Figure 1b]. Figure 2 displays the XRD patterns of UIO−66, its modified structures, and the immobilized enzymes produced by utilizing them as carriers. The diffraction peak distribution pattern of UIO−66 is consistent with previously reported UIO−66 materials, confirming the successful synthesis of UIO−66 [35]. Furthermore, the diffraction peaks of UIO−66/Pro were comparable to those of conventional UIO−66, indicating that the addition of Pro did not alter the crystal structure of UIO−66. The ROL@UIO−66/Pro material, prepared through covalent binding, exhibited an identical diffraction pattern to the conventional UIO−66/Pro material. This suggests that the presence of the enzyme did not induce any alterations in the crystal structure of UIO−66/Pro during the immobilization process using the covalent-binding method [36]. The immobilized enzyme produced by the carrier adsorption method did not exhibit significant diffraction peaks and showed a broadening of the peaks, suggesting a decrease in the crystallization capacity of the immobilized enzyme prepared by adsorption [37].

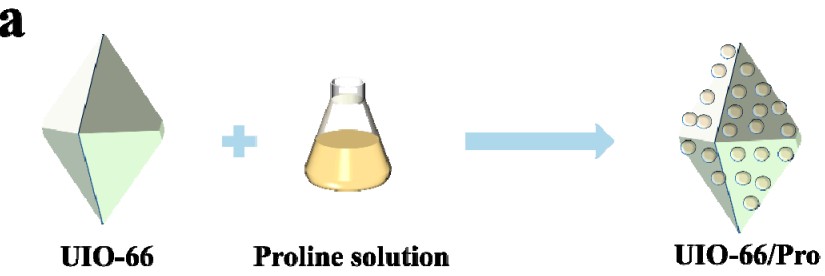

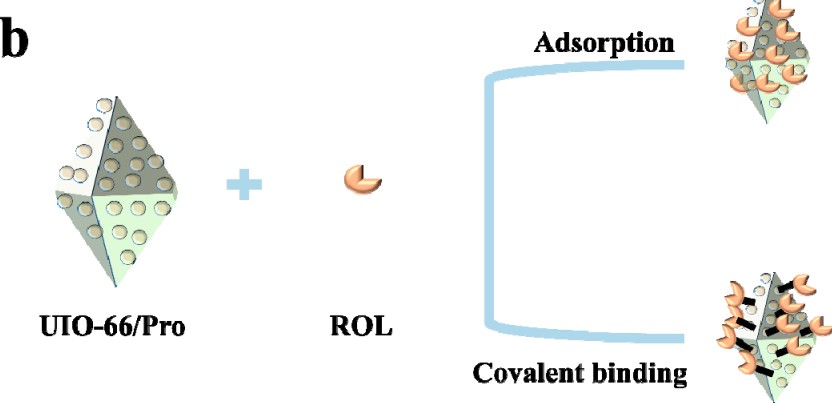

**Figure 1.** Schematic illustration of the preparation process for the ROL@UIO−66 (**a**) and ROL@UIO−66/Pro (**b**).

In addition, SEM images showed that the synthesized UIO−66, UIO−66/Pro, ROL@UIO−66, and ROL@UIO−66/Pro prepared by adsorption and the ROL@UIO−66/Pro prepared by the covalent-binding method possess a distinct crystal structure with well-defined octahedral morphology. Furthermore, the immobilized enzyme prepared using UIO−66/Pro and the covalent-binding method exhibits a slightly smaller size compared to UIO−66. On the other hand, the immobilized enzyme prepared through adsorption has the

smallest size, which aligns with the XRD diffraction peak results [38]. The rhombic shape of UIO−66 can be observed by TEM, and the fuzzy edges of the TEM of the immobilized enzyme are due to the attachment of the enzyme to the carrier surface. EDS analysis reveals the presence of elements C, N, O, and Zr in UIO−66, UIO−66/Pro, ROL@UIO−66, and ROL@UIO−66/Pro, as seen in Figure 3. This confirms the production of UIO−66 and the uniform distribution of the carrier and immobilized enzyme elements (Figure 4). It was also shown by TGA results that the weight loss of the immobilized enzymes ROL@UIO−66 and ROL@UIO−66/Pro occurring between 200 °C and 500 °C was mainly due to thermal degradation of the enzyme protein molecules (Figure 5), providing further evidence for the presence of ROL. The findings of SEM, EDS, and TGA show that ROL can be successfully fixed by the method proposed in this work.

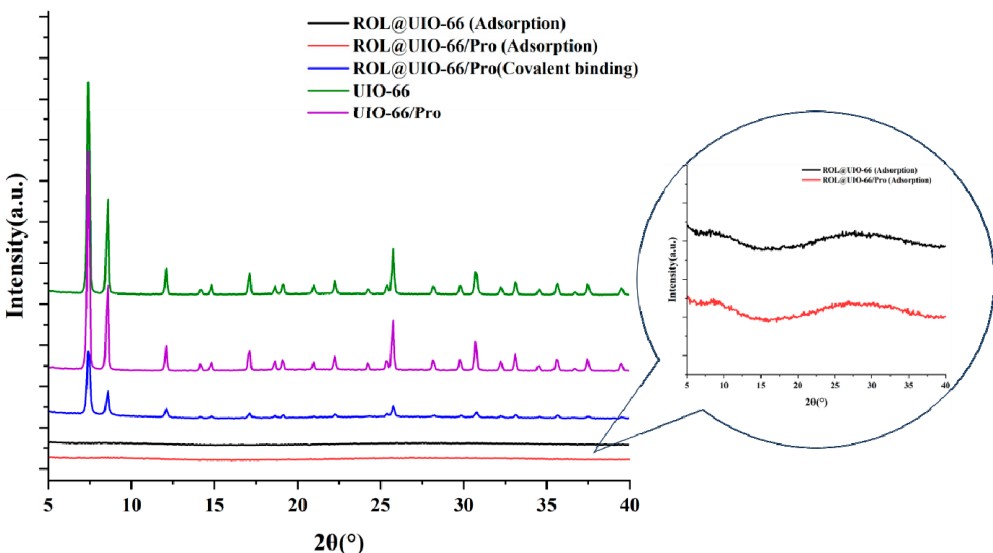

**Figure 2.** XRD patterns of UIO−66, UIO−66/Pro, ROL@UIO−66 (adsorption), ROL@UIO−66/Pro (adsorption), and ROL@UIO−66/Pro (covalent binding).

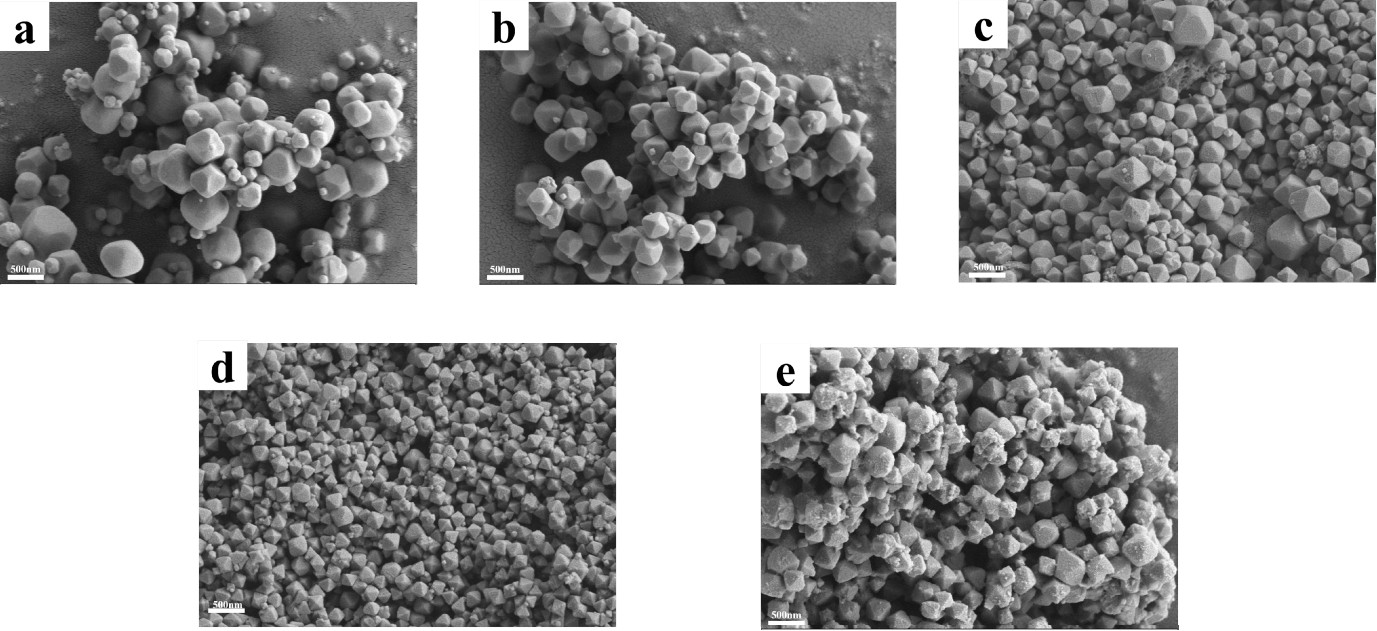

**Figure 3.** SEM of UIO−66 (**a**), UIO−66/Pro (**b**), ROL@UIO−66 by adsorption (**c**), ROL@UIO−66/Pro by adsorption (**d**), and ROL@UIO−66/Pro by covalent binding (**e**).

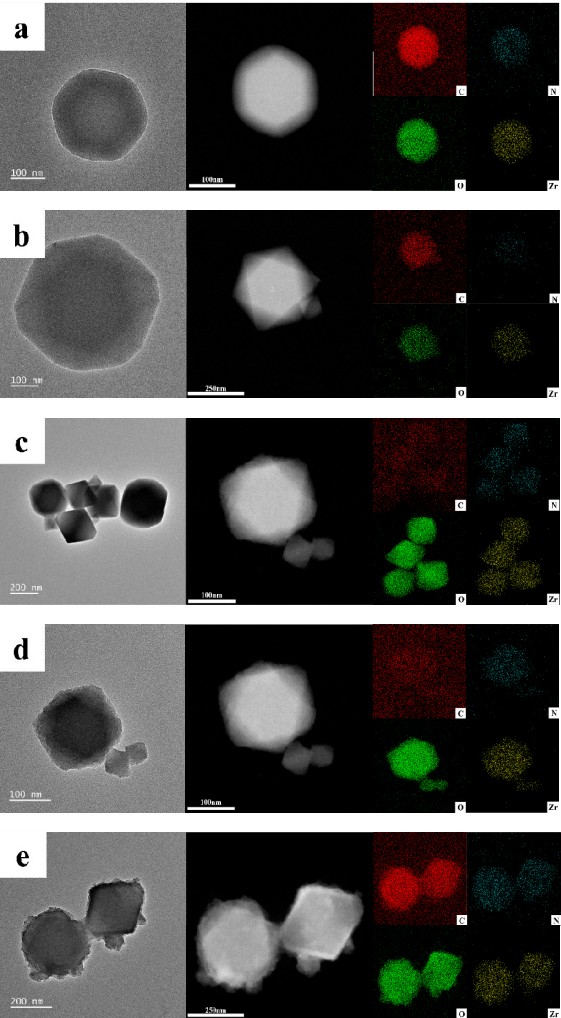

**Figure 4.** TEM of UIO−66 (**a**), UIO−66/Pro (**b**), ROL@UIO−66 by adsorption (**c**), ROL@UIO−66/Pro by adsorption (**d**), and ROL@UIO−66/Pro by covalent binding (**e**).

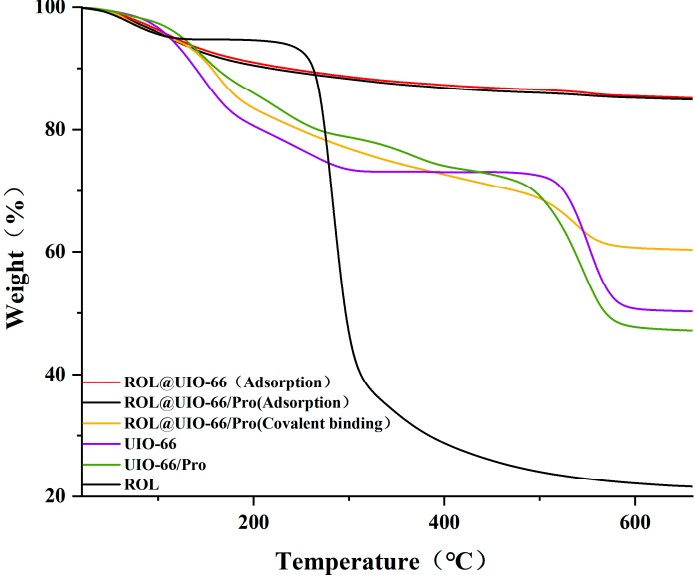

**Figure 5.** TGA of UIO−66, UIO−66/Pro, ROL@UIO−66 (adsorption), ROL@UIO−66/Pro (adsorption), and ROL@UIO−66/Pro (covalent binding).

As seen in Figure 6, UIO−66, UIO−66/Pro, ROL@UIO−66, and ROL@UIO−66/Pro exhibited specific absorption peaks at 655, 1018, 1097, 1156, 1382, 1500, and 1578 cm$^{-1}$. The peak located at 655 cm$^{-1}$ is the stretching vibration peak of ν(Zr-O) in the secondary building block of MOF [39]; the peaks located at 1382, 1500, and 1578 cm$^{-1}$ are the ν(COO-) out-of-plane stretching vibration peaks [40]; and the peaks located at 1018, 1097, and 1156 cm$^{-1}$ are the ν(C-H) vibration peaks of the benzene ring in terephthalic acid [41], indicating the formation of UIO−66.

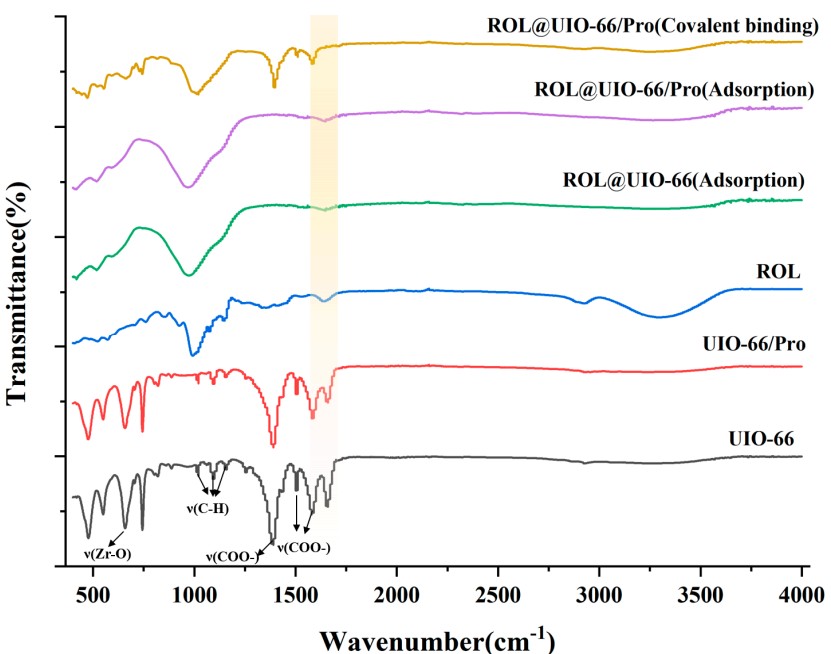

**Figure 6.** Fourier infrared spectroscopy of UIO−66, UIO−66/Pro, ROL@UIO−66 (adsorption), ROL@UIO−66/Pro (adsorption), and ROL@UIO−66/Pro (covalent binding).

Regarding N$_2$ adsorption/desorption isotherms, both the UIO−66 material and UIO−66/Pro material displayed type I adsorption isotherms, indicating their microporous nature. Additionally, the N$_2$ adsorption/desorption isotherms confirmed the presence of a type I isotherm, further confirming the microporous structure of both the UIO−66 material and UIO−66/Pro material. Conversely, the N$_2$ adsorption–desorption isotherms of ROL@UIO−66 and ROL@UIO−66/Pro, which were prepared using the adsorption method, exhibited type-II adsorption isotherm characteristics with a noticeable hysteresis loop (Figure 7c,d). This indicates the presence of both micropores and mesopores in ROL@UIO−66 and ROL@UIO−66/Pro. The N$_2$ adsorption–desorption isotherms of ROL@UIO−66/Pro, synthesized using the covalent cross-linking approach, exhibited type I adsorption isotherm characteristics. This suggests that both ROL@UIO−66 and ROL@UIO−66/Pro possess a microporous structure [42,43]. The specific surface areas of UIO−66 and UIO−66/Pro reached 817.944 m$^2$/g and 513.717 m$^2$/g, suggesting that UIO−66 and UIO−66/Pro could be used as suitable carriers for enzyme immobilization due to their high micropore structure and specific surface area. After pore size calculation by the density functional theory (DFT) method, UIO−66/Pro matched the pore size distribution curve of UIO−66 in the micropore interval [36]. The results showed a decrease in pore volume from 0.399 cc/g (UIO−66) to 0.274 cc/g (UIO−66/Pro) (Table 1). This indicates that the Pro occupies a portion of the pore channel of UIO−66. Furthermore, the UIO−66/Pro exhibited excellent properties as an immobilized carrier and altered the properties of UIO−66 after Pro modification, which was similar to previously reported findings [44,45]. It was reported that there are ROL molecules, a protein molecule with length a = 92.77 Å, b = 128.86 Å, and c = 78.35 Å (from The National Center for Biotechnology Information), which are smaller than those of the carrier. So, it appears that the enzyme blocked some of

the pore channels since the pore volumes of immobilized enzymes obtained by adsorption were 0.020 cc/g (ROL@UIO−66) and 0.023 cc/g (ROL@UIO−66/Pro), which were even less than those of the carrier. The pore volume of the immobilized enzyme (ROL@UIO−66/Pro) created using the covalent-binding approach was larger (0.099 cc/g) than that achieved with the adsorption method. This indicates that the occurrence of pore clogging was minimized, resulting in a decrease in mass transfer resistance [46]. The mesoporous structure observed in the immobilized enzyme, generated via the adsorption approach, is likely a result of structural modifications that occur during the adsorption process. These modifications led to local instability and collapse of the pores in UIO−66, as indicated by the XRD pattern [47,48]. In addition, the hydrophilicity of UIO−66, UIO−66/Pro, and the immobilized enzymes was tested by contact angle measurements. Water droplets on UIO−66 had a contact angle of 79.30° (Figure 8a), which was increased to 120.95° (Figure 8b) after Pro modification, with a significant increase in the hydrophobicity of the surface, which is in agreement with the results of Rani et al. [33]. Following the immobilization of ROL, the contact angles of the immobilized enzymes were all reduced to varying degrees, which may be attributed to the enhanced hydrophilicity of the immobilized enzymes. The increased hydrophobicity (75.59°) of the enzyme ROL@UIO−66/Pro when immobilized through covalent bonding, compared to those immobilized by the adsorption method [ROL@UIO−66 (37.37°) and ROL@UIO−66/Pro (39.70°)], can be attributed to the greater extent of covalent bonding between the enzyme and the carrier surface, rather than to direct plugging of the pores of the carrier. The above results indicate that the hydrophilicity of the immobilized enzyme prepared by the two methods varies greatly.

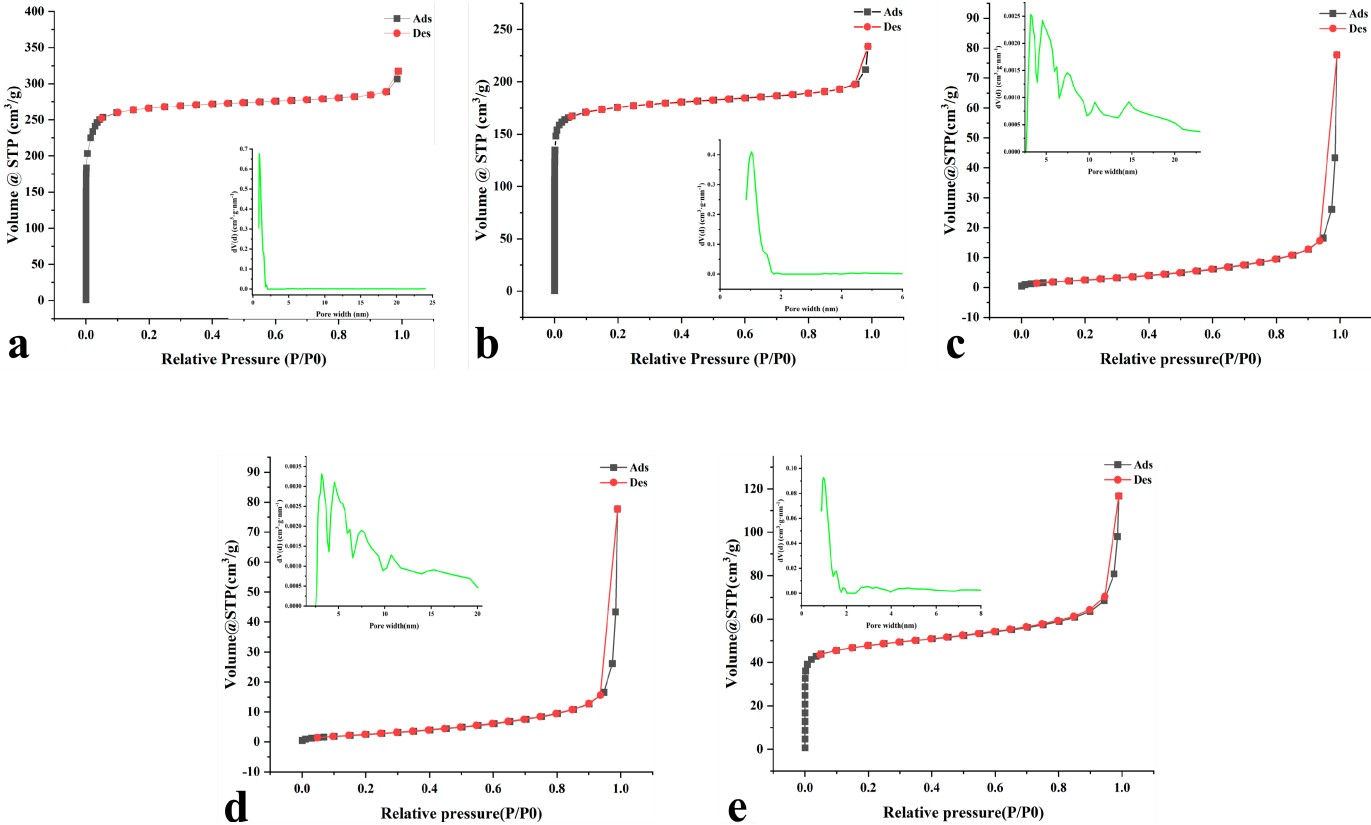

**Figure 7.** Nitrogen sorption isotherms and mesoporous size distributions of UIO−66 (**a**), UIO−66/Pro (**b**), ROL@UIO−66 by adsorption (**c**), ROL@UIO−66/Pro by adsorption (**d**), and ROL@UIO−66/Pro by covalent binding (**e**).

**Table 1.** BET surface area, pore volume, and pore width of UIO−66, UIO−66/Pro, ROL@UIO−66 by adsorption, ROL@UIO−66/Pro by adsorption, and ROL@UIO−66/Pro by covalent binding.

| Biocatalysts | Surface Area/(m$^2$·g$^{-1}$) | | Pore Volume/(cc/g) | | Pore Width (nm) | |
|---|---|---|---|---|---|---|
| | BET | BJH | DFT | BJH | HK | DFT |
| UIO−66 | 817.944 | 0.081 | 0.399 | 3.396 | 0.593 | 0.899 |
| UIO−66/Pro | 513.717 | 0.092 | 0.274 | 3.396 | 0.603 | 1.029 |
| ROL@UIO−66(Adsorption) | 5.313 | 0.073 | 0.020 | 3.388 | 0.368 | 3.169 |
| ROL@UIO−66/Pro (Adsorption) | 6.762 | 0.121 | 0.023 | 3.388 | 0.368 | 3.169 |
| ROL@UIO−66/Pro (Covalent binding) | 160.202 | 0.111 | 0.099 | 3.799 | 0.608 | 0.984 |

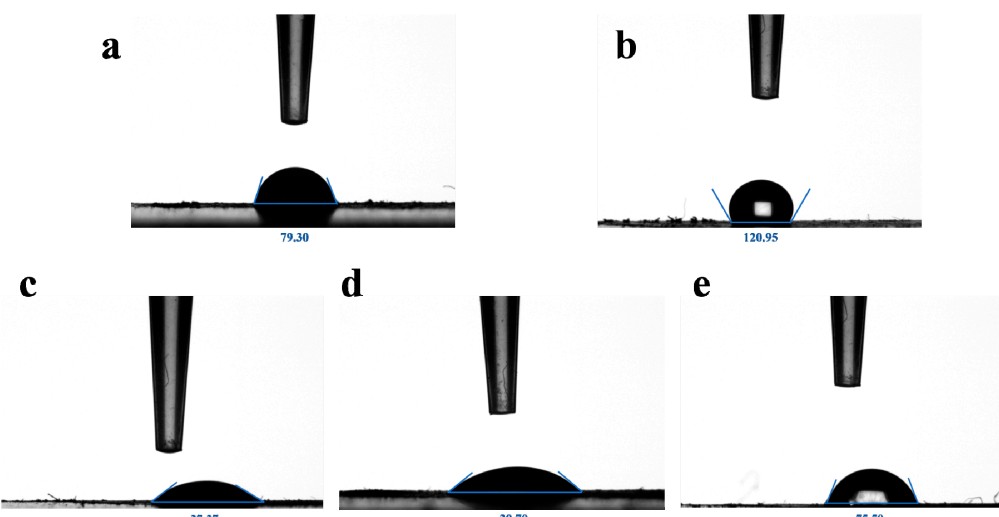

**Figure 8.** Contact angle of UIO−66 (**a**), UIO−66/Pro (**b**), ROL@UIO−66 by adsorption (**c**), ROL@UIO−66/Pro by adsorption (**d**), and ROL@UIO−66/Pro by covalent binding (**e**).

## 2.2. Catalytic Performance of ROL@UIO−66 and ROL@UIO−66/Pro

Figure 9 displays the results of a comparison of the activity of the immobilized enzymes ROL@UIO−66 and ROL@UIO−66/Pro that were produced using free ROL, adsorption, and the covalent-bonding technique. The research conducted revealed that the enzyme activity of the immobilized enzyme, generated using the adsorption method utilizing UIO−66/Pro as the carrier, was approximately 1.5 times more than that of the immobilized enzyme using UIO−66 as the carrier. The immobilized enzyme ROL@UIO−66/Pro, produced using the covalent-bonding technique, exhibited 1.4 times more activity compared to the free enzyme. The presence of the two carriers did not affect the determination of the activity. Comparison of the specific enzyme activities of the immobilized enzymes ROL@UIO−66 and ROL@UIO−66/Pro produced by free ROL, adsorption, and covalent-bonding techniques are detailed in Schedule S1. Among them, the immobilized enzyme prepared by the covalent method showed a superior enhancement of enzyme activity of 0.0647 U/mg, which was 1.73 times higher than that of the free enzyme (0.0373 U/mg). The results indicated that there may be hydrogen bonding between the UIO−66 material and Pro, forming a unique nanopore environment that exhibited greater adsorption capacity [49]. The primary impact of the immobilized enzyme constructed through the covalent-bonding technique is likely due to its ability to effectively maintain the structural conformation of the ROL. Additionally, it could even cause a favorable alteration in the ROL's conformation, leading to an improvement in catalytic activity [50]. During the process of immobilizing enzymes through adsorption, the presence of ROL may block the pores of the carrier material. This blockage leads to an increase in the resistance to mass transfer and a decrease in enzyme activity. Additionally, the ROL that blocks the carrier pores can cause changes in the enzyme's conformation during the immobilization process, thereby affecting its catalytic activity [47].

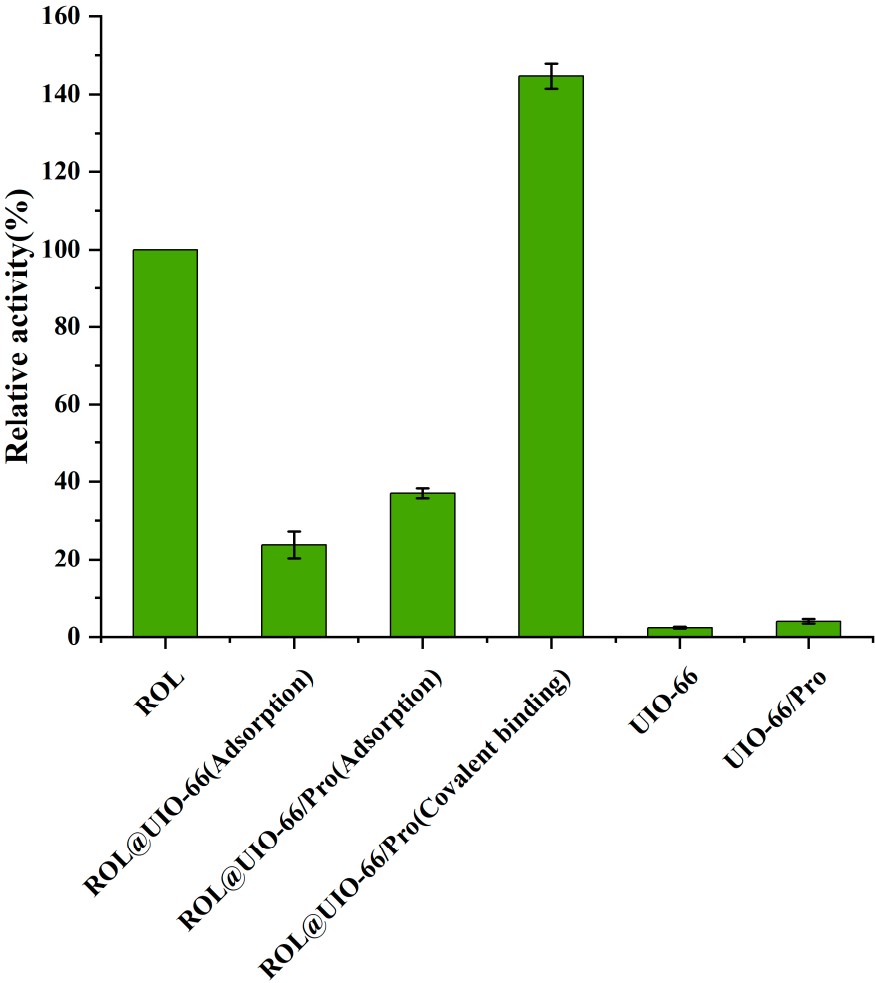

**Figure 9.** Hydrolysis of ROL, ROL@UIO−66 (adsorption), ROL@UIO−66/Pro (adsorption), ROL@UIO−66/Pro (covalent binding), UIO−66, and UIO−66/Pro.

Moreover, kinetic parameters were obtained by nonlinear curve fitting between the immobilized enzyme and the *p*-NPB concentration (Figure 10). The $K_m$ value of the free ROL was 2.1 times higher than that of the ROL@UIO−66/Pro prepared by the covalent-bonding method. This indicates that ROL@UIO−66/Pro has a much stronger affinity towards the substrate compared to free ROL. It suggests that the enzyme's conformation may have altered in a way that promotes an upsurge in substrate affinity. Nevertheless, it is worth noting that the $K_m$ values of the immobilized enzymes ROL@UIO−66 (131.193 mM) and ROL@UIO−66/Pro (121.367 mM) obtained through the adsorption method were 2.59 and 2.39 times higher than those of free ROL (Table 2). This suggests that the substrate affinity of the immobilized enzymes prepared using the adsorption method was weaker compared to that of the free ROL, aligning with the findings of the enzyme activity. In the case of $V_{max}$, the activity of the immobilized enzymes ROL@UIO−66 (0.05966 mM/min) and ROL@UIO−66/Pro (0.0713 mM/min), prepared using the adsorption method, was higher than that of the immobilized enzyme prepared employing the covalent-binding method (0.0265 mM/min) and the free enzyme (0.0312 mM/min). This difference in activity could be attributable to the fact that the immobilized enzyme was generated utilizing the adsorption approach. This could be attributed to the presence of structural defects in the enzyme generated using the adsorption approach, which amplifies the rate of catalytic activity. In contrast, the decrease in $V_{max}$ of the immobilized enzyme prepared by the covalent-binding method was due to the elevated mass transfer resistance caused by the immobilized carrier. The enzyme turnover number (*k*cat) is the catalytic constant of an enzyme and is defined as the amount of substrate that can be converted per active center or

per molecule of enzyme per unit of time. In the calculation of $k$cat, the protein concentration of the enzyme and the magnitude of $V$max are crucial [51]. For immobilized enzymes based on MOFs, the framework structure of the carrier increases the mass transfer resistance of the enzyme, making the $k$cat number of the immobilized enzyme smaller than that of the free enzyme [52]. In addition, $k$cat/$K_m$ is used to reflect both the affinity and the catalytic ability of the enzyme for the substrate, and, thus, can be used for the comparison of the catalytic efficiencies of both the free and immobilized enzymes for specific substrates [53]. The $k$cat/$K_m$ values of ROL@UIO−66/Pro prepared by adsorption were higher than those of ROL@UIO−66 prepared by adsorption, indicating that the catalytic efficiency of ROL@UIO−66/Pro was higher than that of ROL@UIO−66. Moreover, the $k$cat/$K_m$ of ROL@UIO−66/Pro prepared by covalent binding was higher than that of ROL@UIO−66 and ROL@UIO−66/Pro prepared by adsorption, suggesting that the catalytic efficiency of the immobilized enzyme prepared by the covalent-binding method was higher than that of the immobilized enzyme prepared by the adsorption method [34]. However, the value of $k$cat/$K_m$ of the immobilized enzyme was still lower than that of the free enzyme, which may be due to the increase in mass transfer resistance [52].

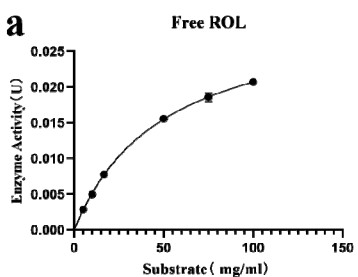
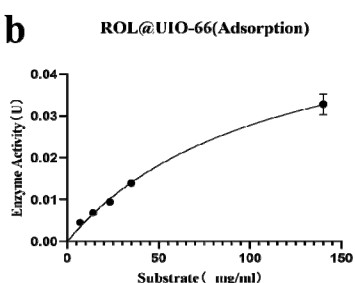
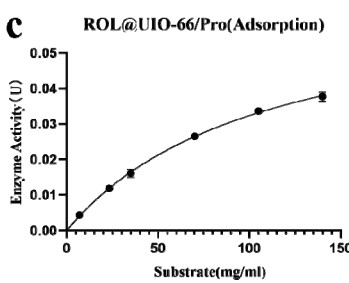
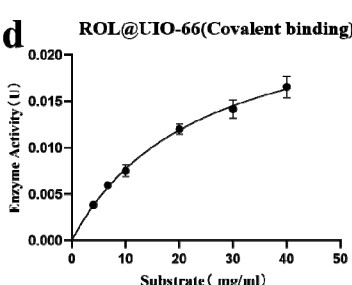

**Figure 10.** The nonlinear kinetic analysis curve of ROL(**a**), ROL@UIO−66 by adsorption (**b**), ROL@UIO−66/Pro by adsorption (**c**), and ROL@UIO−66/Pro by covalent binding (**d**).

**Table 2.** Enzymatic kinetic parameters of ROL, ROL@UIO−66, and ROL@UIO−66/Pro.

| Biocatalysts | $K_m$ (mM) | $V_{max}$ (mM/min) | $V_{max}/K_m$ | $k_{cat}$ (1/s) | $k_{cat}/K_m$ (L·mg$^{-1}$·s$^{-1}$) |
|---|---|---|---|---|---|
| Free ROL | 50.653 ± 2.463 | 0.0312 ± 0.0007 | 0.000616 ± 0.00002 | 0.303 ± 0.00685 | 0.00598 ± 0.000226 |
| ROL@UIO−66(Adsorption) | 131.193 ± 36.188 | 0.05966 ± 0.0109 | 0.000463 ± 0.00005 | 0.0634 ± 0.0116 | 0.000493 ± 0.0000504 |
| ROL@UIO−66/Pro (Adsorption) | 121.367 ± 16.511 | 0.0713 ± 0.00540 | 0.000591 ± 0.00004 | 0.138 ± 0.0104 | 0.00114 ± 0.0000751 |
| ROL@UIO−66/Pro (Covalent binding) | 24.033 ± 5.758 | 0.0265 ± 0.00363 | 0.00112 ± 0.000121 | 0.0835 ± 0.0114 | 0.00354 ± 0.000380 |

### 2.3. The stability of ROL@UIO−66 and ROL@UIO−66/Pro

The temperature stability of free ROL and the immobilized enzymes ROL@UIO−66 and ROL@UIO−66/Pro is shown in Figure 11. Here, the specific enzyme activity value measured after holding at 30 °C is recorded as 100% as a comparison, and the initial specific enzyme activity value is detailed in Schedule S2. The temperature stability of the immobilized enzyme ROL@UIO−66/Pro, synthesized utilizing a covalent-binding technique, exhibited superior performance compared to that of the immobilized lipases

ROL@UIO−66 and ROL@UIO−66/Pro synthesized by the adsorption technique, with a holding time of 1 h at temperatures ranging from 30 to 70 °C. The effect of pH on the enzyme activity of ROL, ROL@UIO−66, and ROL@UIO−66/Pro is shown in Figure 12. Here, the specific enzyme activity value measured after storage at pH 6.5 is recorded as 100% as a comparison, and the initial specific enzyme activity value is detailed in Schedule S2. As seen in Figure 12, the pH stability of the immobilized enzyme ROL@UIO−66/Pro prepared using the covalent-binding method was better than that of the immobilized lipase ROL@UIO−66 and ROL@UIO−66/Pro prepared using the adsorption method. The enzyme activity of ROL@UIO−66/Pro produced by the covalent-binding technique remained higher than that of the free enzyme, even at pH 8.5. It was shown that the immobilized enzyme prepared using covalent binding has better alkali tolerance than the immobilized enzyme prepared using the adsorption method. The results suggest that the immobilized enzyme prepared using the covalent-binding method maintains the optimal temperature and pH for ROL better than the immobilized enzyme prepared using the adsorption method. This indicates that the enzyme's conformation can be preserved to a greater extent, and potentially even altered to enhance enzyme activity.

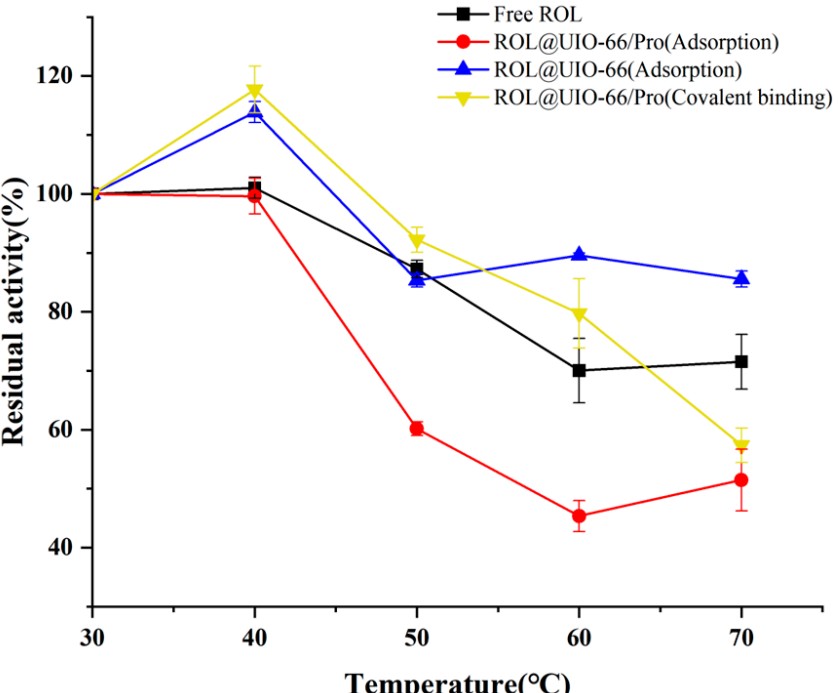

**Figure 11.** Effects of different pH on enzymatic activities of ROL, ROL@UIO−66 (adsorption), ROL@UIO−66/Pro (adsorption), and ROL@UIO−66/Pro (covalent binding).

Figure 13 illustrates the reusability of the immobilized enzyme by establishing the initial conversion rate as 100% in the first cycle and utilizing it as a reference to calculate the enzyme's reusability. Here, the specific enzyme activity value measured after the first cycle is recorded as 100% as a comparison, and the initial specific enzyme activity value is detailed in Schedule S2. In the comparison of reusability, the adsorption method of the immobilized enzyme has better reusability than the covalent-binding method of the immobilized enzyme. However, in the ninth cycle, the enzyme activity of the covalent-binding method was still higher than that of the immobilized enzyme prepared by the adsorption method (Table 3). This phenomenon can be attributed to the blocking of carrier pores by ROL during the preparation of immobilized enzymes using the adsorption method. As a result, the carrier provides lower initial enzyme activity but offers protection to the enzyme. On the other hand, the immobilized enzyme prepared using the covalent-binding method results in better enzyme conformation but reduces enzyme protection.

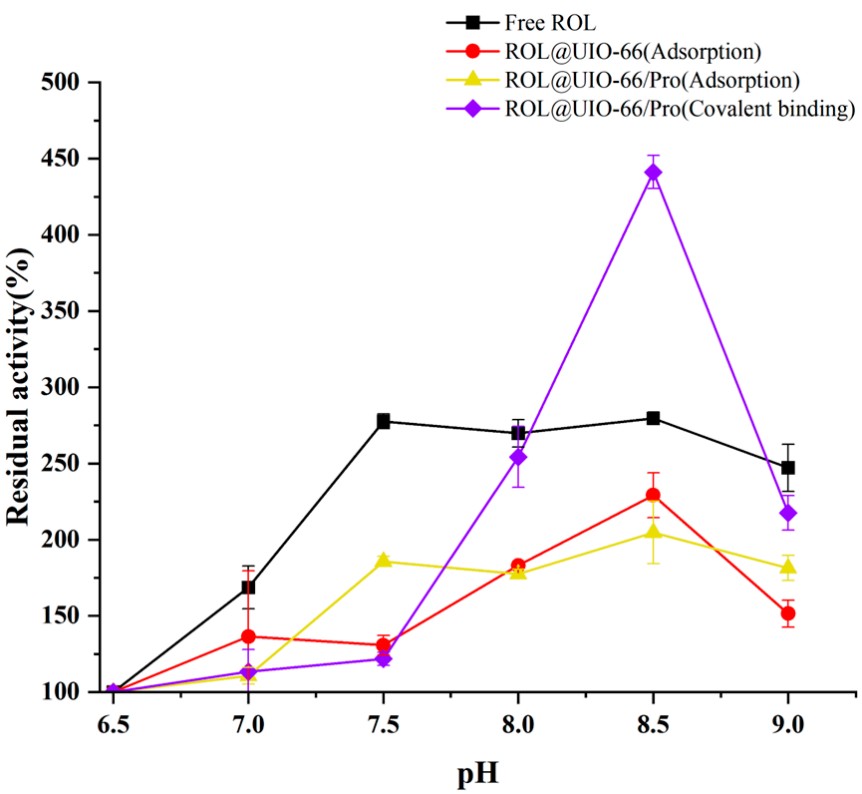

**Figure 12.** Effects of different temperature on enzymatic activities of ROL, ROL@UIO−66 (adsorption), ROL@UIO−66/Pro (adsorption), and ROL@UIO−66/Pro (covalent binding).

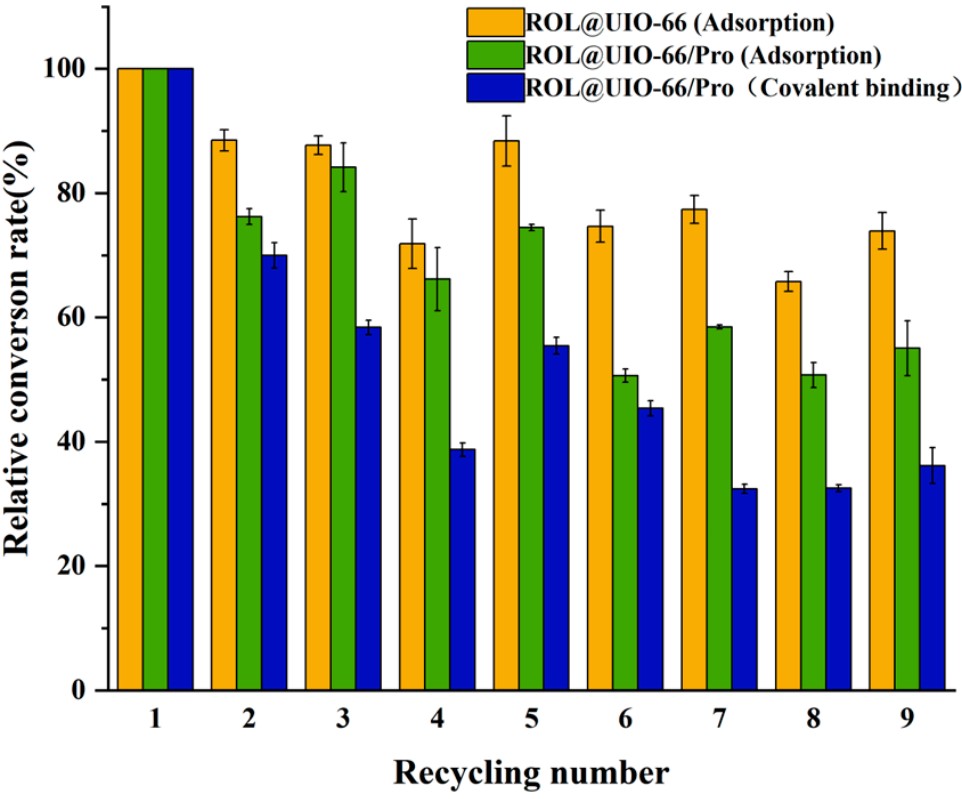

**Figure 13.** Relative conversion rate of ROL@UIO−66 (adsorption), ROL@UIO−66/Pro (adsorption), ROL@UIO−66 (covalent binding), and ROL@UIO−66/Pro (covalent binding).

**Table 3.** Comparison of enzyme activity after nine cycles.

| | ROL@UIO−66 (Adsorption) | ROL@UIO−66/Pro (Adsorption) | ROL@UIO−66/Pro (Covalent Binding) |
|---|---|---|---|
| Enzyme activity(U) | 0.00270 | 0.00275 | 0.00431 |

## 3. Materials and Methods

### 3.1. Materials

N, N-dimethylformamide (DMF) (99.5%), L-Pro (99%), ROL, P-nitrophenyl butyrate (*p*-NPB, ≥98%), and p-nitrophenol (*p*-NP) were purchased from Sigma-Aldrich (Louis, MO, United States). Acetic acid, sodium dihydrogen phosphate (99.5%), ammonium sulfate, and disodium hydrogen phosphate (99%) were purchased from Tianjin Fuchen Chemical Reagent Factory (Tianjin, China). Zirconium (IV) chloride (≥99.9%), glutaric dialdehyde (25%), terephthalic acid (99%), and isopropyl alcohol (>99%) were provided by Aladdin Reagent Inc. (Shanghai, China). Methanol (≥99.5%) and ethanol (99.5%) were purchased from Shanghai McLean Biochemical Technology (Shanghai, China).

### 3.2. Synthesis of UIO−66 and UIO−66/Pro

Referring to the method of Cheng et al. [33], ZrCl4 (334 mg) and CH3COOH (7 mL) were completely dissolved in DMF (50 mL) under sonication conditions. Then, terephthalic acid (250 mg) dissolved in DMF (50 mL) was added to the above solution (sealed 250 mL sampling bottles were placed together). The resulting mixture was sonicated for 10 min and then stirred for 24 h at 120 °C, 300 rpm in an electrically heated oil bath. After cooling to room temperature, UIO−66 crystals were obtained by centrifugation (10,000 rpm, 15 min) and repeated washing with DMF.

Pro powder (1 g) was dissolved in ethanol (80 mL) and heated at 50 °C for 1 h with continuous stirring (sealed 250 mL sampling bottles were placed together). The heated Pro solution was added to 400 mg of UIO−66 and stirred for 24 h at 50 °C, 300 rpm in an electrically heated oil bath. After being cooled to room temperature, centrifugation (10,000 rpm, 10 min) was followed by several washes with ethanol and methanol. In the end, UIO−66/Pro crystals were vacuum-dried for 12 h at 100 °C to obtain them.

### 3.3. Immobilization of ROL

#### 3.3.1. Adsorption

An amount of 10 mg of UIO−66 or UIO−66/Pro was solubilized in 15 mL of 1 mg/mL ROL solution in a 50 mL centrifuge tube. The mixture was then sonicated in an ultrasonic mixer-washer for 2 min and stirred on a magnetic stirrer at 120 rpm for 12 h at room temperature. The immobilized enzyme was obtained after washing twice with 5 mL of PBS (0.05 M, pH = 7.5).

#### 3.3.2. Covalent Binding

Referring to Hu et al., 50 mg of UIO−66 or UIO−66/Pro was combined with 2 mL of 2.5 mg/mL enzyme dissolved in the enzyme. After stirring on a magnetic stirrer for 2 h, 100% saturated ammonium sulfate solution was added. The cross-linking process required 30 min of stirring at 4 °C, the addition of 25% glutaric dialdehyde, and 3 h of stirring at 4 °C. Following immobilization, the enzyme immobilized on ROL@UIO−66/Pro was obtained after washing three times with 5 mL of PBS (0.05 M, pH = 7.5) [27].

### 3.4. Lipasae Activity Assay and Stability

The catalytic activity and stability of ROL@UIO−66 and ROL@UIO−66/Pro were evaluated based on *p*-NPB hydrolysis. The unit of enzyme activity was defined as the release of 1 mM *p*-NP per minute per milligram of immobilized lipase.

The substrate solutions were prepared as follows: substrate solution A: 2 mg/mL of *p*-NPB solution (isopropanol as solvent) and substrate solution B: 0.05 M pH 7.5 PBS

buffer containing 10% Triton X-100, A:B volume ratio (1:9). After weighing the prepared immobilized enzyme, 1 mL of 2 mg/mL substrate solution and 1 mL of 0.05 mol/L (pH 7.5) PBS buffer were added, and the reaction was carried out at 150 rpm for 5 min in a water bath shaker. At the end of the reaction, the centrifuge tube was quickly placed on ice, the supernatant was collected by centrifugation at 6000 rpm at 4 °C, and the absorbance value was measured at 405 nm. The *p*-NP concentration was determined from the *p*-NP standard curve (Y = 7.5544X + 0.0511, $R^2$ = 0.9998). Reactions were performed in triplicate. Control experiments were performed by replacing fixed ROL with a free ROL solution. Determination of protein concentration of free and immobilized enzymes using Bradford protein assay kit (Thermo Fischer Scientific, Kandel, Germany), and the specific enzyme activity was calculated from the protein concentration.

### 3.5. Kinetics Analysis

The kinetic parameters of the lipase and immobilized lipase samples were determined by assaying ROL, ROL@UIO−66, and ROL@UIO−66/Pro enzyme activities using different concentration gradients of *p*-NPB as substrate to determine the initial reaction rates. $K_m$ and $V_{max}$ values were calculated from the Michaelis–Menten equation according to GraphPad Prism 8 software.

### 3.6. Characterization

After the samples were freeze-dried using a vacuum freeze dryer (Free zone 4.5 plus, Labconco, Kansas City, MO, USA), the morphological characteristics of MOFs, modified MOFs, and immobilized enzyme were observed using a jsm7610f scanning electron microscope (SEM) and a jem2100p transmission electron microscope (TEM), (JEOL, Akishima City, Tokyo, Japan) to observe the morphological characteristics of MOFs, modified MOFs, and immobilized enzymes. X-ray diffraction (XRD) patterns were obtained with a D8advance model X-ray diffractometer, Bruker, Pileeka, MA, USA; pore sizes were obtained using N2 adsorption–desorption apparatus, Quantachrome, Boynton Beach, FL, USA; scanning transmission electron microscopy-energy dispersive X-ray spectrometry (STEM-EDS) was used to obtain the elemental distribution of the samples (JEOL, Akishima City, Tokyo, Japan); and a Nicolet IS50 Fourier transform infrared spectroscopy (FT-IR) was used to collect in the range of 400–4000 cm$^{-1}$(Thermo Fisher Scientific, Waltham, MA, USA).

### 3.7. Stability Measurements

The enzyme activities of ROL, ROL@UIO−66, and ROL@UIO−66/Pro were determined by placing them in 0.05 mol/L PBS buffer at different pH values. The samples were kept at room temperature for 60 min.

ROL, ROL@UIO−66, and ROL@UIO−66/Pro were incubated in 0.05 mol/L PBS buffer at pH 7.5 for 60 min at various temperatures to determine their enzyme activities.

For ROL, ROL@UIO−66, and ROL@UIO−66/Pro, the hydrolytic stability of the immobilized lipases was assessed using nine sequential hydrolysis reactions. After each ester hydrolysis reaction, the immobilized lipases were collected by centrifugation (6000 rpm, 5 min) and washed twice with 5 mL of PBS solution (0.05 M, pH = 7.5) in order to remove residual substrate and product before the next cycle. The conversion rate for the initial cycle was 100%, and the relative conversion rate is calculated as the ratio of the remaining conversion rate to the maximum conversion rate for each sample.

## 4. Conclusions

On the whole, the enzyme system in this study was constructed by modifying UIO−66 via Pro, which served as a soft template derived from UIO−66 as a precursor, in accordance with the previously established methodology. When comparing immobilized enzymes generated through adsorption and covalent-binding methods, it is shown that the covalent-crosslinking method has superior properties in terms of enzyme immobilization. To the best of our knowledge, this is the first time that an immobilized enzyme has been prepared

using UIO−66/Pro as a carrier using the covalent-binding method. This concept facilitates the advancement of biotechnological applications of post-synthesis modified MOFs for immobilizing lipase using various immobilization methods.

**Supplementary Materials:** The following supporting information can be downloaded at: https://www.mdpi.com/article/10.3390/catal14030180/s1, Schedule S1: Comparison of the free ROL with the immobilized enzyme; Schedule S2 Enzyme stability measures initial enzyme activity.

**Author Contributions:** Conceptualization, X.L.; methodology, C.Z.; software, X.D.; formal analysis, X.D., C.Z. and W.L.; investigation, X.D.; data curation, X.D.; writing—original draft preparation, X.D.; writing—review and editing, C.Z. and P.J.P.; visualization, X.D.; supervision, X.L. All authors have read and agreed to the published version of the manuscript.

**Funding:** This research was funded by the National Natural Science Foundation of China (No. 31830069, 32001638).

**Data Availability Statement:** The data that support the findings of this study are available from the corresponding author upon reasonable request.

**Acknowledgments:** We thank Kaile Zhao for explaining morphological characteristics of MOFs.

**Conflicts of Interest:** The authors declare no conflicts of interest.

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
