# Peer review of "Preparation of Proline-Modified UIO−66 Nanomaterials and Investigation of Their Potential in Lipase Immobilization"

_catalysts, doi:10.3390/catal14030180_

Round 1
Reviewer 1 Report
Comments and Suggestions for Authors
The manuscript “catalysts-2880006” was reviewed. Although this study is interesting to the readers of this journal, I recommend some suggestions.
- Lines 18, 35 and 43 (Page 1): Rhizopus oryzae in itallic. I recommend revise all manuscript.
- Lines 20/22: The authors should add technical information (in brief) on the immobilization process and catalytic activity or immobilized protein amount values.
- Introduction: Expalin better why Rhizopus oryzae lipase was chosen to peform this study. I recommend report biotechnological applications of this enzyme in industrial processes. I recommend cite some references:
https://doi.org/10.1016/j.plipres.2016.08.001
https://doi.org/10.1016/j.ijfoodmicro.2017.06.012
https://doi.org/10.1007/s11274-022-03465-4
- Results: The authors should firstly introduce in this section the immobilization parameters of each heterogeneous biocatalyst such as immobilized protein amound, hydrolytic or esterification activities, and, finally, specific activity (ratio between activity and immobilized protein amount). These results should be compared with free lipase to demonstrate the effect of immobilization protocol on the properties of the enzyme.
- Lines 113/114: The authors should better explain why the cristalinity of the support prepared was influenced by the immobilization method. I recommend revise this sentence.
- Table 1: The authors should compare the molecular diameter of Rhizopus oryzae lipase with the pore size values of each support. This could contribute with the discussion (preferential immobilization on the internal and/or external support surface and mass transfer effects on the catalytic efficiency of the different heterogeneus biocatalysts prepared in this study). The introduction of proline on the MOF struture and introduction of fucntional groups can also reduce the pore size of the support. I recommend that these results is compared with the catalytic performance of free lipase.
- Figures 9, 11, 12 and 13: Provide experimental values of catalytic activity for “100% relative activity” for each biocatalyst in Figure legend.
- I recommend that the authors explain (in detail) the experimental apparatus utilized in Sections 3.2 and 3.3. I suggest report initial volume, reactor dimensions, temperature control method, stirrer type (mechanical or magnetic), intensity and impeller configuration used in this study.
- Section 3.5: The authors should better describe the experimental conditions used in this study. (initial substrate concentration values, temperature, pH and ionic strenght, initial hydrolytic activity values for each biocatalyst – free and immobilized lipase).
Comments on the Quality of English LanguageThe manuscript “catalysts-2880006” was reviewed. Although this study is interesting to the readers of this journal, I recommend some suggestions.
- Lines 18, 35 and 43 (Page 1): Rhizopus oryzae in itallic. I recommend revise all manuscript.
- Lines 20/22: The authors should add technical information (in brief) on the immobilization process and catalytic activity or immobilized protein amount values.
- Introduction: Expalin better why Rhizopus oryzae lipase was chosen to peform this study. I recommend report biotechnological applications of this enzyme in industrial processes. I recommend cite some references:
https://doi.org/10.1016/j.plipres.2016.08.001
https://doi.org/10.1016/j.ijfoodmicro.2017.06.012
https://doi.org/10.1007/s11274-022-03465-4
- Results: The authors should firstly introduce in this section the immobilization parameters of each heterogeneous biocatalyst such as immobilized protein amound, hydrolytic or esterification activities, and, finally, specific activity (ratio between activity and immobilized protein amount). These results should be compared with free lipase to demonstrate the effect of immobilization protocol on the properties of the enzyme.
- Lines 113/114: The authors should better explain why the cristalinity of the support prepared was influenced by the immobilization method. I recommend revise this sentence.
- Table 1: The authors should compare the molecular diameter of Rhizopus oryzae lipase with the pore size values of each support. This could contribute with the discussion (preferential immobilization on the internal and/or external support surface and mass transfer effects on the catalytic efficiency of the different heterogeneus biocatalysts prepared in this study). The introduction of proline on the MOF struture and introduction of fucntional groups can also reduce the pore size of the support. I recommend that these results is compared with the catalytic performance of free lipase.
- Figures 9, 11, 12 and 13: Provide experimental values of catalytic activity for “100% relative activity” for each biocatalyst in Figure legend.
- I recommend that the authors explain (in detail) the experimental apparatus utilized in Sections 3.2 and 3.3. I suggest report initial volume, reactor dimensions, temperature control method, stirrer type (mechanical or magnetic), intensity and impeller configuration used in this study.
- Section 3.5: The authors should better describe the experimental conditions used in this study. (initial substrate concentration values, temperature, pH and ionic strenght, initial hydrolytic activity values for each biocatalyst – free and immobilized lipase).
Author Response
The manuscript “catalysts-2880006” was reviewed. Although this study is interesting to the readers of this journal, I recommend some suggestions.
Response: Thank you for your comments.
- Lines 18, 35 and 43 (Page 1): Rhizopus oryzae in itallic. I recommend revise all manuscript.
Response: Thank you for your careful work. We have corrected the sentence according to your careful suggestion, and the revision has been highlighted by using blue colored text. (p1, line 18,35 and 43)
- Lines 20/22: The authors should add technical information (in brief) on the immobilization process and catalytic activity or immobilized protein amount values.
Response: Thank you for your thoughtful comment. We have modified the relevant section to briefly describe the specific enzyme activity of the immobilized enzyme ROL@UIO-66/Pro (p1, lines 20-22).
- Introduction: Expalin better why Rhizopus oryzae lipase was chosen to peform this study. I recommend report biotechnological applications of this enzyme in industrial processes. I recommend cite some references:
https://doi.org/10.1016/j.plipres.2016.08.001
https://doi.org/10.1016/j.ijfoodmicro.2017.06.012
https://doi.org/10.1007/s11274-022-03465-4
Response: Thank you for your suggestions. We have added the application of the relevant ROL in industrial production, the application literature is labeled as [4] and [16], and the changes made are marked in blue font. (p2, lines 50-55)
- Results: The authors should firstly introduce in this section the immobilization parameters of each heterogeneous biocatalyst such as immobilized protein amound, hydrolytic or esterification activities, and, finally, specific activity (ratio between activity and immobilized protein amount). These results should be compared with free lipase to demonstrate the effect of immobilization protocol on the properties of the enzyme.
Response: Thank you for your suggestions. As a matter of fact, because of the change of morphology during the immobilization of the enzyme, it is preferred to analyze the carrier and the immobilized enzyme by the change of morphology. Therefore, we preferred to analyze the morphological structure of the carrier and the immobilized enzyme in order to analyze the change of enzyme activity by the change of morphological structure. Thank you for your careful reading of the article, the data and description of the relevant enzyme activity have been added in the article, and will be explained to you in detail in the following answer.
- Lines 113/114: The authors should better explain why the cristalinity of the support prepared was influenced by the immobilization method. I recommend revise this sentence.
Response: Thank you for your suggestions. In fact, we believe that different immobilized enzyme methods change their crystallinity in correlation with the length of immobilization and carrier pore collapse at the time of immobilization, and we have revised the relevant sentence and marked it in blue font (p3, lines 116-121).
- Table 1: The authors should compare the molecular diameter of Rhizopus oryzae lipase with the pore size values of each support. This could contribute with the discussion (preferential immobilization on the internal and/or external support surface and mass transfer effects on the catalytic efficiency of the different heterogeneus biocatalysts prepared in this study). The introduction of proline on the MOF struture and introduction of fucntional groups can also reduce the pore size of the support. I recommend that these results is compared with the catalytic performance of free lipase.
Response: Thank you for your thoughtful comment. We made changes in the appropriate places and found the size data for ROL from the NCBI database and labeled it using blue font (p7, lines 184-186). The collapse of the carrier pore size caused by the adsorption method and the change of enzyme entry into the carrier pore and thus enzyme activity compared to free enzyme are presented in the enzyme activity analysis (p9, lines 233-243).
- Figures 9, 11, 12 and 13: Provide experimental values of catalytic activity for “100% relative activity” for each biocatalyst in Figure legend.
Response: Thank you for your suggestions. Regarding the relevant data, we have compiled them as Exhibit 2. We present them to you here:
Schedule 2 Enzyme stability measures initial enzyme activity
Biocatalysts |
Free ROL |
ROL@UIO-66(Adsorption) |
ROL@UIO-66/Pro(Adsorption) |
ROL@UIO-66/Pro(Covalent binding) |
Temperature stability |
||||
Specific activity(U/mg) |
0.0265±0.00269 |
0.00244±0.000472 |
0.00823±0.00081 |
0.0631±0.00454 |
pH stability |
||||
Specific activity(U/mg) |
0.0198±0.00727 |
0.00137±0.000593 |
0.00361±0.000137 |
0.00415±0.000543 |
Relative conversion rate |
||||
Specific activity(U/mg) |
- |
0.00372±0.000393 |
0.0104±0.000214 |
0.0665±0.00116 |
- I recommend that the authors explain (in detail) the experimental apparatus utilized in Sections 3.2 and 3.3. I suggest report initial volume, reactor dimensions, temperature control method, stirrer type (mechanical or magnetic), intensity and impeller configuration used in this study.
Response: Thank you for your suggestions. We have made changes in the Methods section and labeled them in blue (p14, lines 335-338 and 341-342; p15, lines 343-344, 349-352 and 354-356).
- Section 3.5: The authors should better describe the experimental conditions used in this study. (initial substrate concentration values, temperature, pH and ionic strenght, initial hydrolytic activity values for each biocatalyst – free and immobilized lipase).
Response: Thank you for your suggestions. We have modified the methods section, the enzymatic reaction kinetics experiment section is only for enzyme activity determination at different concentrations of substrate. The method of enzyme activity determination is consistent with the above description (p15, lines 377-379).
Reviewer 2 Report
Comments and Suggestions for Authors
The manuscript submitted by Dong et al. describes the preparation and characterization of physicochemical and biochemical characterizations of lipase immobilized on proline-modified UIO-66 nanomaterial. The manuscript is generally sound, and I have no concerns about the nanomaterial's preparation part and physicochemical characterization. However, the biochemical part and its execution raise my objections (listed below). Therefore, I recommend a major revision.
Major issue:
The authors indicated that the immobilized enzyme ROL@UIO-66/Pro, produced with the covalent-bonding technique, exhibited greater activity (about 1.73 times that of the free enzyme). On the other hand, no information about concentrations of the free enzyme and immobilized one is provided. Different KM and Vmax values probably result from a different number of catalytic centers (in other words, active enzyme molecules in a solution for a free enzyme and immobilized on nanoparticles). The authors must estimate enzyme concentrations (or average distribution on nanoparticles) and calculate kcat values to compare these different enzyme systems. It allows them to calculate the catalytic efficiency (kcat/Km), an absolute value used to compare various enzyme systems.
Minor editing of English language required.
Author Response
The manuscript submitted by Dong et al. describes the preparation and characterization of physicochemical and biochemical characterizations of lipase immobilized on proline-modified UIO-66 nanomaterial. The manuscript is generally sound, and I have no concerns about the nanomaterial's preparation part and physicochemical characterization. However, the biochemical part and its execution raise my objections (listed below). Therefore, I recommend a major revision.
Response: Thank you for your comments.
Major issue:
The authors indicated that the immobilized enzyme ROL@UIO-66/Pro, produced with the covalent-bonding technique, exhibited greater activity (about 1.73 times that of the free enzyme). On the other hand, no information about concentrations of the free enzyme and immobilized one is provided. Different KM and Vmax values probably result from a different number of catalytic centers (in other words, active enzyme molecules in a solution for a free enzyme and immobilized on nanoparticles). The authors must estimate enzyme concentrations (or average distribution on nanoparticles) and calculate kcat values to compare these different enzyme systems. It allows them to calculate the catalytic efficiency (kcat/Km), an absolute value used to compare various enzyme systems.
Response: Thank you for your careful work. We have checked and modified the article as follows: the data on the specific enzyme activities of the immobilized enzymes added are shown in Exhibit 1 which has been modified in blue font (p9, lines 228-232); the method section of the immobilized enzyme preparation process has been modified by adding a section on enzyme concentration (p15, lines 349-352 and 354-356); and data related to the kcat and kcat/Vmax of the free enzyme and immobilized enzyme were added to Table 2, and the related analyses were modified using blue font (p11, lines 266-272). enzyme kcat and kcat/Vmax related data were added to Table 2, and the related analysis was modified using blue font (p11, lines 266-272). Finally, thank you again for your valuable suggestions on the article.
Reviewer 3 Report
Comments and Suggestions for Authors
The focus of the manuscript is on the preparation of zirconium-based metal oxide framework materials and their application as supports for lipases. The topic is of current interest to researchers in materials science, biotechnology, and biocatalysis. The aims and objectives are clearly stated. The introduction is informative and focused. The description of the methods is sufficiently detailed and the choice of methods is appropriate. They are appropriately chosen. The methodology is relevant to the study. The results are clearly presented. They are well illustrated with thirteen figures and three tables. The results are analyzed. A logical conclusion is drawn.
My recommendation is for acceptance of the paper in its present form.
Author Response
The focus of the manuscript is on the preparation of zirconium-based metal oxide framework materials and their application as supports for lipases. The topic is of current interest to researchers in materials science, biotechnology, and biocatalysis. The aims and objectives are clearly stated. The introduction is informative and focused. The description of the methods is sufficiently detailed and the choice of methods is appropriate. They are appropriately chosen. The methodology is relevant to the study. The results are clearly presented. They are well illustrated with thirteen figures and three tables. The results are analyzed. A logical conclusion is drawn.
My recommendation is for acceptance of the paper in its present form.
Response: Thank you for your comments.
Round 2
Reviewer 1 Report
Comments and Suggestions for Authors
The manuscript was corrected, as suggested. I still recommend a carefull revision of all references before publication.
Comments on the Quality of English LanguageThe manuscript was corrected, as suggested. I still recommend a carefull revision of all references before publication.
Author Response
The manuscript was corrected, as suggested. I still recommend a carefull revision of all references before publication.
Response: Thank you for your comments. We've checked all the references.
Reviewer 2 Report
Comments and Suggestions for Authors
The current version of the manuscript is slightly improved. However, there are still some issues that need to be clarified.
- There needs to be more consistency in the text. On the one hand, the authors indicate that “…the immobilized enzyme 228 prepared by the covalent method showed a superior enhancement of enzyme activity of 229 0.0647 U/mg, which was 1.73 times higher than that of the free enzyme (0.0373 U/mg)” (page 9, line 228-230, and abstract). On the other hand, kcat and kcat/Km values (Table 2) indicate the opposite: Free ROL is significantly more active than immobilized enzymes. The authors should clarify it.
- Also, significant digits for kcat, kcat/Km and their standard deviations should be unified.
Comments on the Quality of English Language
Minor editing of English language required
Author Response
- There needs to be more consistency in the text. On the one hand, the authors indicate that “…the immobilized enzyme 228 prepared by the covalent method showed a superior enhancement of enzyme activity of 229 0.0647 U/mg, which was 1.73 times higher than that of the free enzyme (0.0373 U/mg)” (page 9, line 228-230, and abstract). On the other hand, kcat and kcat/Km values (Table 2) indicate the opposite: Free ROL is significantly more active than immobilized enzymes. The authors should clarify it.
Response: Thank you for your comments. The article has supplemented and explained the questions you raised, and has been marked in yellow. (p9, lines 264-283)
- Also, significant digits for kcat, kcat/Km and their standard deviations should be unified.
Response: Thank you for your comments. The significant numbers of kcat, kcat/Km and their standard deviations have been unified in the article and are shown in Table 2 using yellow font. (p11, Table 2)